# Meteo and hydrodynamic data in the Mar Grande and Mar Piccolo by the LIC Survey, winter and summer 2015

Michele Mossa[1,2], Elvira Armenio[3], Mouldi Ben Meftah[1,2], Maria Francesca Bruno[1,2], Diana De Padova[1,2], Francesca De Serio[1,2]

[1]Department of Civil, Environmental, Land, Building Engineering and Chemistry (DICATECh), Polytechnic University of Bari, Via Orabona 4 – 70125 Bari – Italy
[2]CoNISMa, Inter University Consortium for Marine Sciences, Piazzale Flaminio 9 – 00196 Rome, Italy
[3]Regional Agency for Environmental Protection ARPA Puglia, Corso Trieste 27 - 70126 Bari - Italy

*Correspondence to*: Diana De Padova (diana.depadova@poliba.it)

**Abstract.** The Coastal Engineering Laboratory (LIC) of the DICATECh of the Polytechnic University of Bari (Italy) maintains a place-based research program in the Mar Grande and Mar Piccolo of Taranto (a coastal system in southern Italy), providing records of hydrodynamic and water-quality measurements. This site is one of the most complex marine ecosystem models in terms of ecological, social, and economic activities. It is considered highly vulnerable for the presence of the naval base, of the biggest refinery of Europe and of the oil refinery. Two fixed stations have been installed, one in the Mar Grande (MG station) and another in Mar Piccolo (MP station). In the MG station constituents include wind speed and direction, air temperature and humidity, barometric pressure, net solar radiation, water salinity, water temperature, water pressure, dissolved oxygen, fluorescence, turbidity, CDOM, crude oil and refined fuels, sea currents and waves. In the MP station constituents include water temperature, sea currents and waves. We provide a summary of how these data have been collected by the research group and how they can be used to deepen understanding of the hydrodynamic structures and characteristics of the basin.

These data are available at http://doi.org/10.5281/zenodo.4449641 (Mossa et al., 2020)

| Design type(s) | time series design |
| --- | --- |
| Measurement Type(s) | MG station:<br>wind speed and direction, air temperature and humidity, barometric pressure, net solar radiation, water salinity, water temperature, water pressure, dissolved oxygen, fluorescence, turbidity, CDOM, crude oil and refined fuels, sea currents and wave.<br><br>MP station:<br>water temperature, sea currents and waves |
| Technology Type(s) | data acquisition system |
| Factor Type(s) | spatiotemporal_interval |
| Sample Characteristic(s) | Mar Grande and Mar Piccolo of Taranto, coastal waters |

# 1 Background & Summary

Coastal sites with typical lagoon features are extremely vulnerable, often suffering scarce circulation (de Swart and Zimmerman, 2009; De Pascalis et al., 2016; De Serio and Mossa, 2016a; De Serio and Mossa, 2016b; Armenio et al., 2016, 2017; De Padova et al., 2020; Carlucci et al., 2020). The two bays of the Mar Piccolo have been considered as two different

ecosystems influencing each other. The Mar Piccolo with its typical lagoon features is extremely vulnerable and is characterized by continue diffusion of contaminants with a strong ecological risk towards the marine ecosystem and human health.

Especially in this case of shallow basins subjected to strong anthropization and urban discharges, it is fundamental to monitor their hydrodynamics and water quality (De Carolis et al., 2013; De Serio and Mossa, 2014; De Serio and Mossa, 2015; De

Serio and Mossa, 2016a; De Padova et al., 2017a; De Padova et al., 2017b; Armenio et al., 2018a-b; De Serio and Mossa, 2018; Armenio et al., 2019; De Serio et al., 2020; Chimienti et al., 2020).

This monitoring action has proved to be a necessary tool for local authorities and stakeholders, allowing to deepen the knowledge of the physical processes recurring in the target basin and to check its real-time status. Moreover, it allows to control sediment transport and effluent discharges, which are all phenomena strictly linked to current magnitudes and

directions (De Serio and Mossa, 2013; Green and Coco, 2014; Ben Meftah et al., 2014; Ben Meftah et al., 2015; Mossa et al., 2017, Ben Meftah et al., 2018; De Serio and Mossa, 2016c).

Therefore, coastal management plans and in situ decision-making should include such monitoring actions to guarantee a thorough knowledge of hydrodynamic and tracers diffusion processes. Generally numerical models are preferred to this scope, because they allow to reproduce and predict marine physical phenomena in relatively short time, with accuracy and with

moderate costs (De Serio et al., 2007; Monti and Leuzzi, 2010; Samaras et al., 2016; Di Bernardino et al., 2016; De Serio et al., 2020). Predictive operational oceanography commonly uses models covering regional, sub-regional and shelf-coastal scales. To study local scales, with resolution of few hundred meters, multiscale modelling systems based on a multiple-nesting approach have been implemented lately (Lane et al., 2009; Federico et al., 2017). Therefore, a large dataset is essential to calibrate and validate modelling systems providing forecasts (Lesser et al., 2004; Korotenko et al., 2010; Sánchez-Arcilla et

al., 2014).

At the same time, a large dataset allows to deduce information on the evolutionary state of the analyzed basin. The present note aims to show how long-term and continuous recordings of meteorological, hydrodynamic and water quality data collected in a semi-enclosed sea can be managed to rapidly provide fundamental insights on its hydrodynamic structure and environmental health. The acquired signals have been analyzed in both time and frequency domain, filtered and grouped in

classes with homogeneous features, then correlated. This simple and repeatable procedure has been applied with good results (De Serio and Mossa, 2015; De Serio and Mossa, 2016a; Armenio et al., 2017a-b), interesting in a predictive perspective and for numerical modelling (Kjerfve and Magill, 1989; Babu et al. 2005; Ferrarin et al., 2008; De Serio and Mossa, 2016c; Benetazzo et al., 2012). Although the typical trends in the water circulation and exchanges have been studied by numerous

models developed for the seas of Taranto, more observations, monitoring actions and numerical modelling are still necessary to better understand the most significant hydrodynamic–biological variability of this coastal basin. The results of these study can be applied for similar zones.

## 2 Method and sampling

The hydrodynamics and water-quality studies in the Mar Grande and Mar Piccolo of Taranto (Fig. 1) of the LIC (Coastal Engineering Laboratory of the DICATECh of the Polytechnic University of Bari) include two fixed stations briefly described below.

In December 2013, the first meteo-oceanographic station MG was mounted in the Mar Grande basin, at the geographical coordinates 40°27.6' N and 17°12.9' E (Figs. 1 and 2). It was funded by the Italian National Project PON R&C 2007-2013 "Magna Grecia" and the RITMARE flagship project (Italian Research for the Sea, where we operated as the research unit Co.N.I.S.Ma.-Polytechnic University of Bari) provided by the Italian Ministry of Education, University and Research (De Serio and Mossa, 2016a; De Serio and Mossa, 2016b; Armenio et al., 2017). The local depth in this station is on average equal to 23.25m. In this station a bottom mounted Acoustic Doppler Current Profiler (ADCP), a multidirectional wave array, a weather station and a CTD (SeaBird SBE-37SIP-ODO with sensors of conductivity, temperature, pressure and dissolved oxygen) were installed (Fig. 2).

Furtherly, sensors by Wet Labs and Turner Design to detect water physical and biochemical parameters completed the station. The water parameters were measured a depth on average equal to 5.6m below the sea surface.

In detail, the weather system by Met Pack was installed on the seamark where the monitoring MG station is present. It records speed and wind direction by means of an ultrasonic sensor. Hourly-averaged values of wind speed and direction are provided with an accuracy of ±2% of the velocity value and ±3° of the direction.

The ADCP (by Teledyne RDI) measured the 3D velocity of currents along the vertical. It uses a Janus configuration consisting of four acoustic beams, paired in orthogonal planes, where each beam is inclined at a fixed angle of 20° to the vertical. The ADCP is bottom mounted, upward facing and has a pressure sensor for measuring mean water depth. The transducer head is at 0.50m above the seafloor. Velocities are sampled along the water column with 0.50m vertical bin resolution and a 1.60m blanking distance. Therefore, the water column is investigated from a distance from the sea bottom z=2.1m up to the most superficial bin not biased by waves. The surface layer, with a thickness on average equal to 2.0m, is excluded from the analysis, to filter out the possible noise in the measurements as well as the wave contribution to currents. Mean current velocity profiles are collected continuously at 1-hour intervals, using an average of 60 measurements acquired every 10s. In this way, hourly-averaged velocity components along the water column are available (De Serio and Mossa, 2016a; De Serio and Mossa, 2016b; Armenio et al., 2017). Figure 3 shows an example of polar plots of the measured bottom and surface currents in January 2015.

In May 2014, funded by the Flagship Project RITMARE, also the station MP was placed in the target area. Namely it was installed in the Navigable Channel, at the geographical coordinates 40.473° N and 17.235° E, because it is the main exchange

between the Mar Piccolo and Mar Grande (Fig. 1). As shown by previous studies and field data analysis from MP station (Armenio et al., 2017a; De Serio and Mossa, 2018; De Serio et al., 2020), the net flow is inflowing at deeper layers, from the bottom up to 4m depth, while in the most superficial layer it is directed outward of the basin. Therefore, monitoring actions, as that of MP station, is necessary to better understand the most significant hydrodynamic–biological variability of this extremely vulnerable coastal site.

It is equipped with a bottom mounted ADCP and a wave array (by Teledyne RDI). The local depth in this station is on average equal to 13.7m. Also in this case, considering the ADCP size and its blanking distance, the current velocities are assessed along the vertical starting from z=2.1m from the sea bed, at constant intervals of 0.5m, up to the most detectable unbiased bin, at z=12.6 m. The acoustic frequency of both the installed ADCPs is 600KHz and their velocity accuracy is 0.3% of the water velocity ±0.003m/s.

In both MG and MP stations, the ADCP measures the component of velocity projected along the beam axis, averaged over a range cell. The cross-spectra between velocities measured at various range cells (either beam to beam or along each beam) contain information about wave direction [2-4]. In other words, each depth cell of the ADCP can be considered an independent sensor that makes a measurement of one component of the wave field velocity. The ensemble of depth cells along the four beams constitutes an array of sensors from which magnitude and directional information about the wave field can be determined.

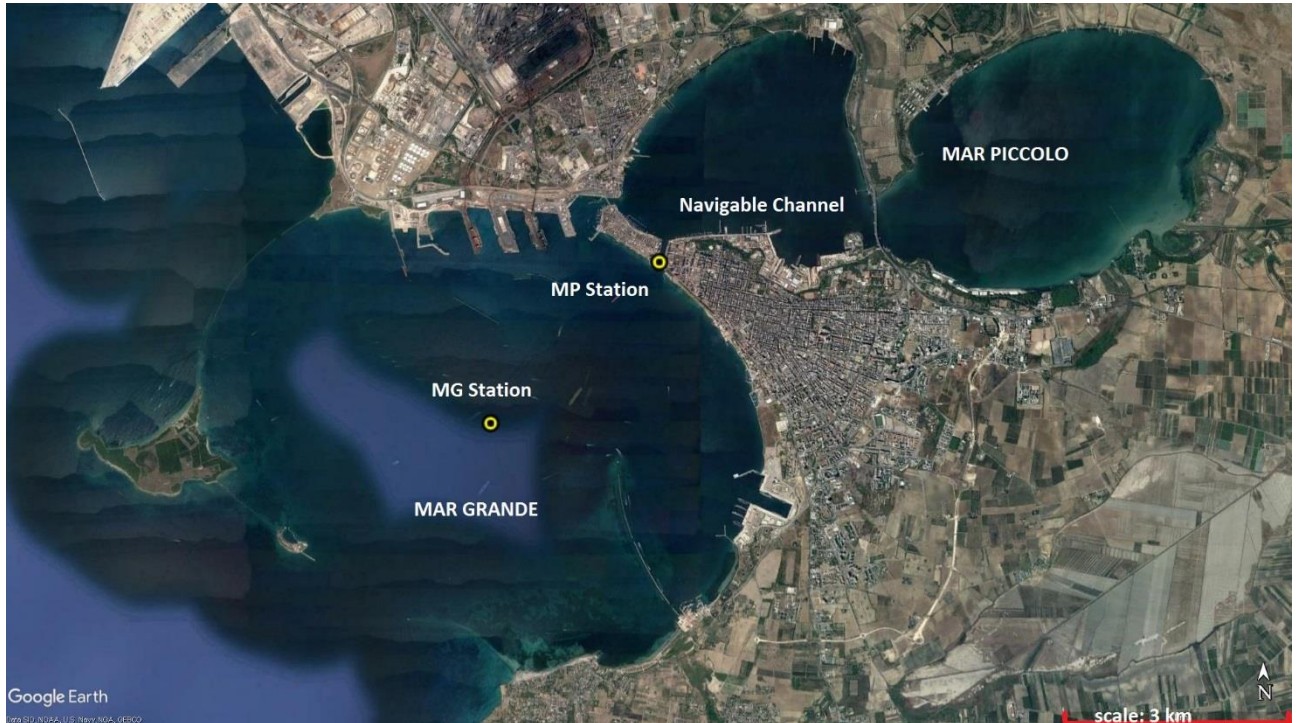

**Figure 1: Map of Mar Grande and Mar Piccolo coastal system, with location of the two monitoring stations MG and MP. Source Google Earth.**

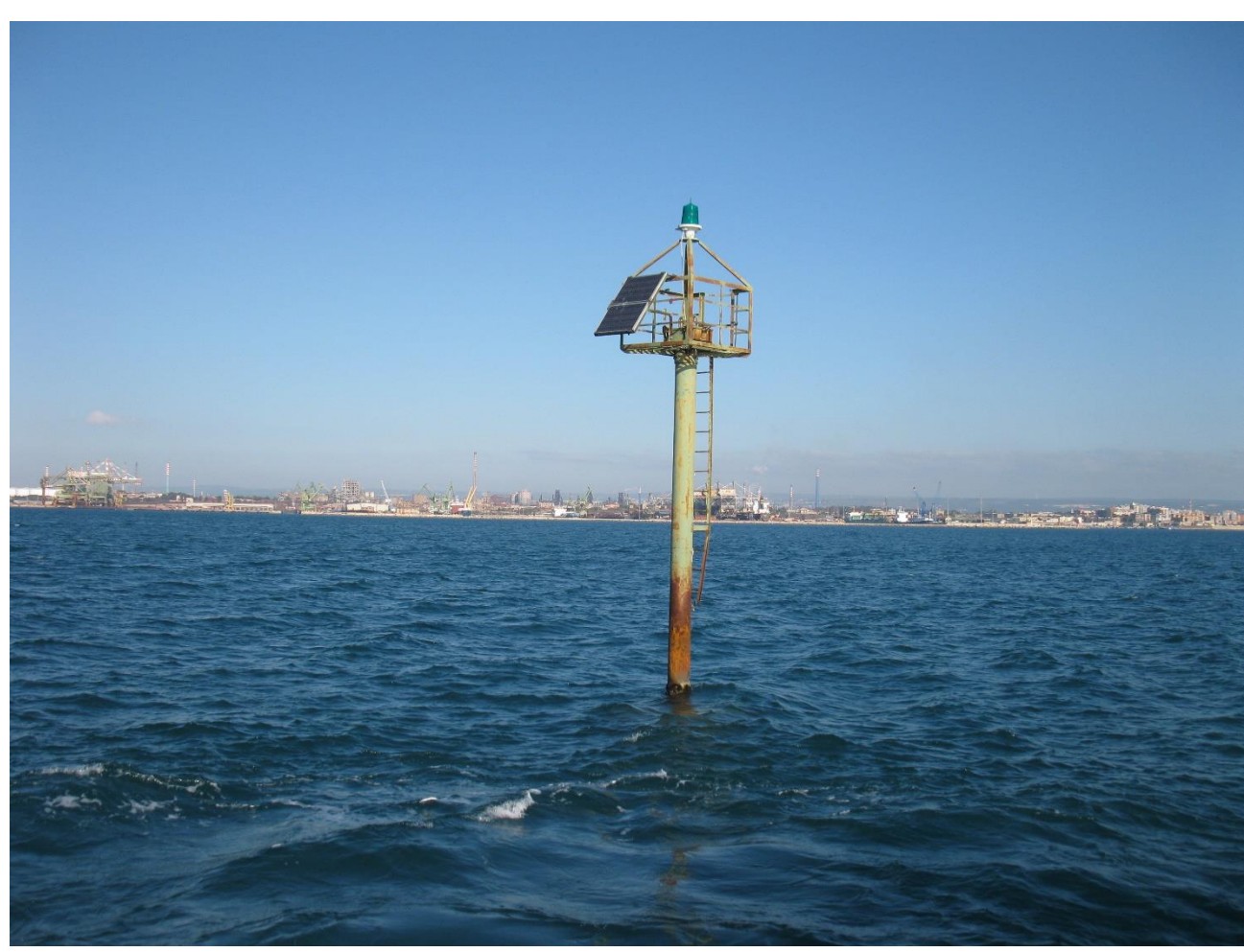

**Figure 2: Seamark where the monitoring station in Mar Grande was installed.**

 **3 Data Record**

The dataset supplied in tab-delimited text format ASCII, contains timeseries of relevant meteocean variables marked up with the SeaDataNet common vocabularies from Library P01, P02 and P03 (https://vocab.seadatanet.org/search vocabularies P01, P02, P03) and divided as follows:

120

| | | Mare Piccolo (MP) | Mare Grande (MG) |
|---|---|---|---|
| **Wave** | Progressive data number Date (year/month/day/hour/minute) Position (Lat, lon) | x | x |
| | | SDN: P01: Conceptid:: GTHDAP01: Significant wave height $H_s$ (m) - SDN: P01: Conceptid:: GTZHAW01: Significant wave period $T_s$ (s) - SDN: P01: Conceptid:: GWMDAD01: Significant wave incoming direction (in degree, referenced to North) - SDN: P01: Conceptid:: MBANZZZZ:Local depth (mm); - SDN: P01: Conceptid:: GTDHAP01: $H_{1/10}$- Average of the 1/10 highest waves - SDN: P01: Conceptid:: GTAMZD01: Average wave period $T_{mean}$ (s) | |
| **Current** | Progressive data number Date (year/month/day/hour/minute) Position (Lat, lon) | x | x |
| | | - SDN: P01: Conceptid:: MBANZZZZ: Cell of measurement with indication of its depth from surface (z=0) - SDN: P01: Conceptid:: LCSAAP01: cell current intensity (m/s) - SDN: P01: Conceptid:: LCDAAP01: cell current direction (in degree, referenced to North) | |
| **Temperature** | Progressive data number Date (year/month/day/hour/minute) Position (Lat, lon) | x | |
| | | - SDN: P01: Conceptid:: MBANZZZZ: Sensor depth (m) - SDN: P01: Conceptid:: TEMPS901: Water Potential Temperature measured in ITS-90 degrees Celsius (°C) | |
| **Meteo** | Progressive data number Date (year/month/day/hour/minute) Position (Lat, lon) | | x |
| | | - SDN: P01: Conceptid:: EGTSSS01: Average wind velocity (m/s) - SDN: P01: Conceptid:: ESSAMX01: Max wind velocity (m/s) - SDN: P01: Conceptid:: EGTDSS01: Wind incoming direction N (deg) - SDN: P01: Conceptid:: CDTAZZ01: Air temperature (°C) - SDN: P01: Conceptid:: CDEWZZ01:Dew point (°C) | |

| | | | x |
|---|---|---|---|
| | | - SDN: P01: Conceptid:: CAPHZZ01:Atmospheric pressure (mbar)<br>- SDN: P01: Conceptid:: CHUMZZ01: Relative humidity (%) | |
| **Water quality** | Progressive data number<br>Date (year/month/day/hour/minute)<br>Position (Lat, lon) | - SDN: P01: Conceptid:: TEMPS901: Water Potential Temperature measured in ITS-90 degrees Celsius (°C)<br>- SDN: P01:Conceptid:: CNDCST01: Conductivity (S/m)<br>- SDN: P01: Conceptid:: PRESPR01: Absolute Pressure (dbar)<br>- SDN: P01 :Conceptid:: PSLTZZ01: Practical Salinity (PSU) using PSS78 algorithm<br>- SDN: P01:Conceptid:: SIGTPR01: Density (kg/m$^3$)<br>- SDN: P01: Conceptid:: DOXYOP01:Dissolved oxygen (ml/l)<br>- SDN: P01: Conceptid:: CLSDPM01: Chlorophyll (µg/l)<br>- SDN: P01: Conceptid:: CLSDPM01: Turbidity (NTU)<br>- SDN: P01: Conceptid:: GP001:CDOM (RFU)<br>- SDN: P01: Conceptid:: GP001: Crude oil (RFU)<br>- SDN: P01: Conceptid:: GP001: Refined oil (RFU) | |

125    The data set has been processed with quality control procedures and data flagged following SeaDataNet protocols.

In particular, the dataset quality control has been carried out with:

- Maintenance and calibration of instruments twice a year in specialized laboratories;
- Visual inspection of the time series (e.g. time series plot, current vector scatter plot, progressive vector diagram, etc.);
130    - Screening together of related parameters such as current speed and current direction or salinity and temperature to identify spurious values;
- Flag spikes in the data;
- Flag suspicious data or correct the data after consultation with the data supplier;
- Check against other data collected on nearby moorings or measured during monitoring survey using two Vessel-
135    Mounted Acoustic Doppler Current Profilers (VM-ADCPs).

## 4 Technical and Data Validation

Results from each sampling data were examined carefully by the research team of the LIC to ensure that all values fell within expected ranges, to verify that calibration regressions were an acceptable basis for computing quantities from sensor measurements and to ensure completeness of each monthly acquisition.

When data were bad acquisitions or were lacking, they were eliminated from the file record.

Each data set was validated with: (1) tests to ensure that the measured values fell within ranges that are plausible and consistent with knowledge of the Mar Piccolo and Mar Grande systems; (2) pattern tests of time series of all measurements to ensure they followed plausible and understandable patterns of variability over time (Babu et al., 2005; Ferrarin et al., 2008).

As also shown in previous works (De Serio and Mossa, 2016a; De Serio and Mossa, 2016b; Armenio et al., 2017), examples of plots deduced by the dataset are displayed below, for MG Station and MP Station.

Figure 3 shows the pattern of the bottom and surface currents typical of the January month. The prevalence of currents directed towards SW confirms the deductions of other experimental works, proving that the site topography controls bed circulation and induces a bottom current outflowing from the SW opening in the Mar Grande border (De Serio and Mossa, 2016a; De Serio and Mossa, 2016b; Armenio et al., 2017) also in accordance with results of numerical models (De Pascalis et al., 2016). On the contrary, the significant waves enter from the same opening and spread throughout, thus not influenced by prevailing winds (Figure 4). Therefore, this SW opening represents a dominant key factor in the hydrodynamic of the basin.

Figure 5 displays the water temperature trend in July month, which increases as expected, and the salinity trend, which seems consistent with increasing evaporation rates and reduced riverine inputs. In Figure 6 the timeseries of dissolved oxygen and chlorophyll are plotted, with peaks due to the algal bloom of the period.

Figures 7a -7b show the time series of the significant wave heights recorded in January 2015 and July 2015, respectively. In January month, the greatest value of observed $H_s$ is equal to 1.2 m, while the average value of $H_s$ is around 0.2 m. In July month, the greatest value of observed $H_s$ is equal to 0.5 m, while the average value of $H_s$ is around 0.15 m.

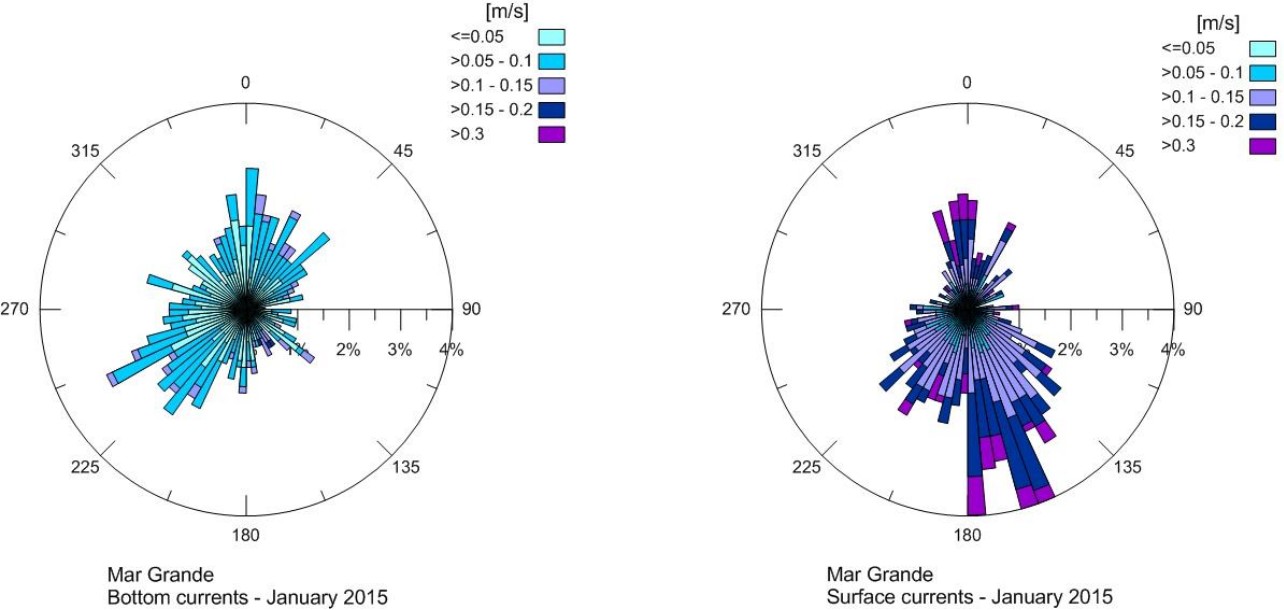

**Figure 3: Polar plot of measured bottom and surface currents in January 2015 (direction of propagation shown).**


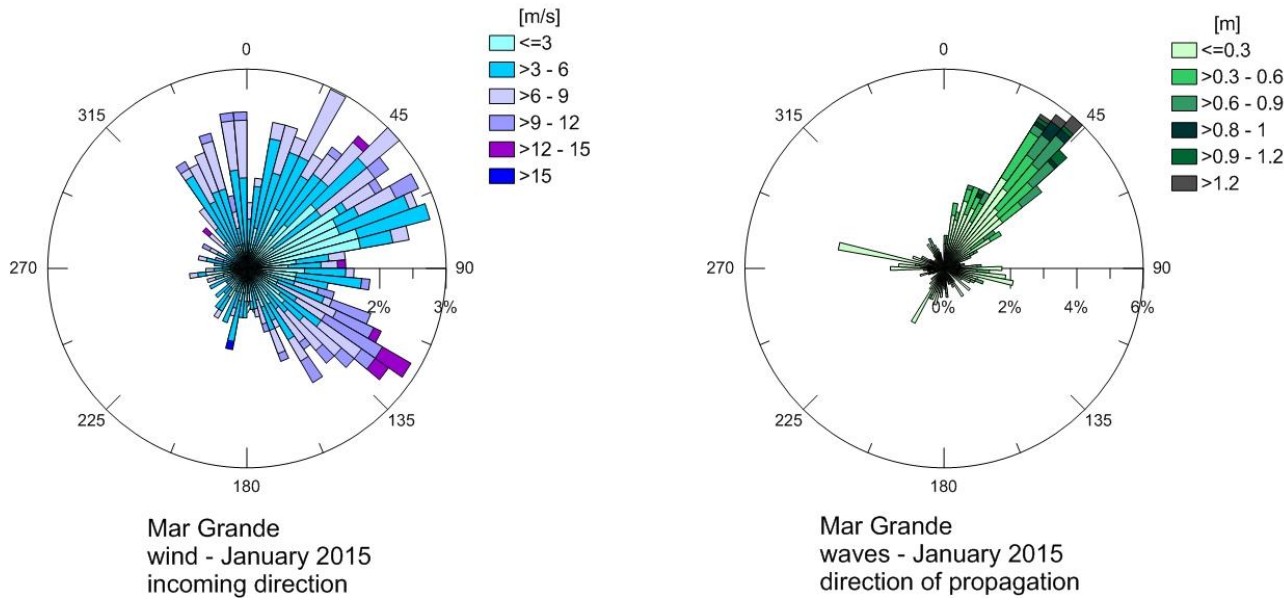

**Figure 4: Polar plot of measured wind and waves in January 2015.**


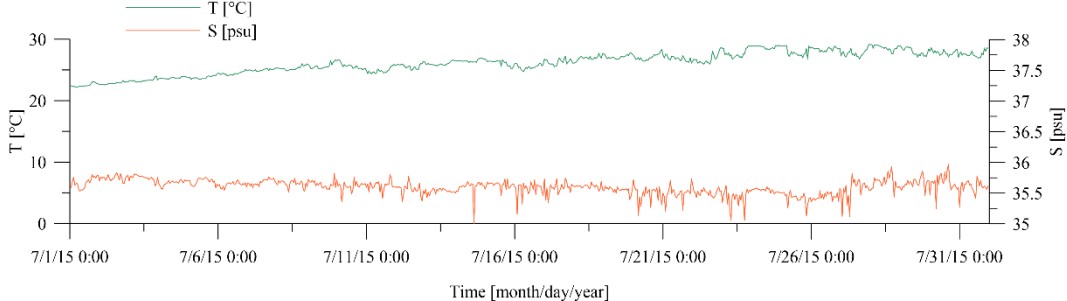

**Figure 5: Time series of measured water temperature [°C] and salinity [psu] in July 2015.**

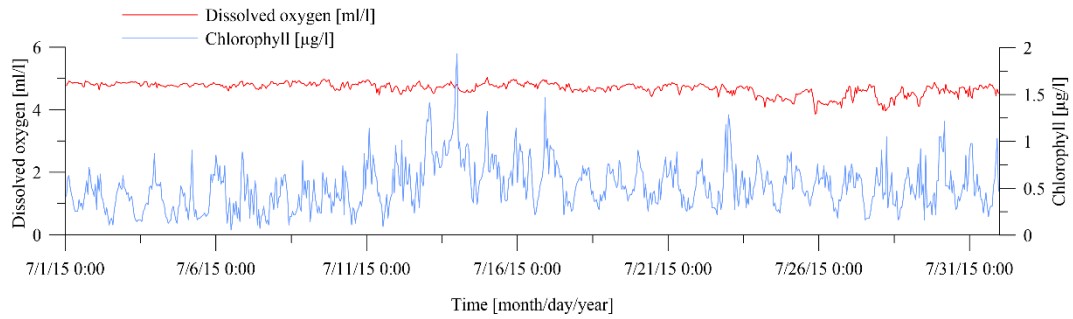

**Figure 6: Time series of measured dissolved oxygen [ml/l] and chlorophyll [μg/l] in July 2015.**

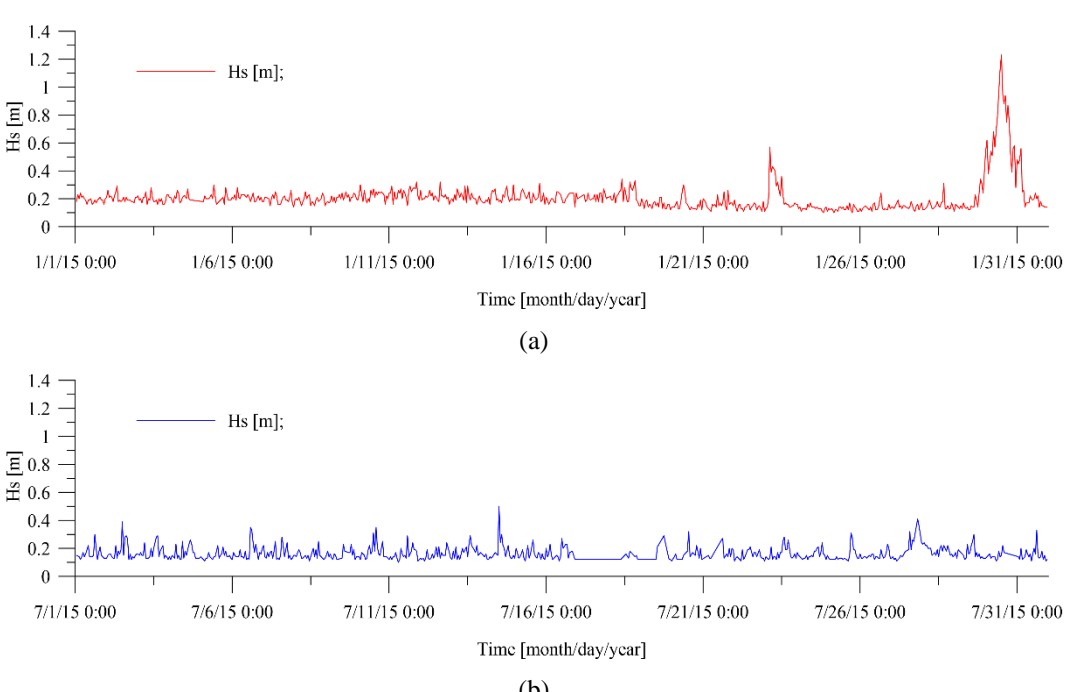

(a)

(b)

**Figure 7: Time series of measured significant wave heights Hs recorded in (a) January 2015 and (b) July 2015.**

## 5 Data availability

All data used in this paper are available at: http://doi.org/10.5281/zenodo.4449641 (Mossa et al., 2020)

## Author Contributions

M.M., E.A., M.B.M., D.D.P., M.F.B. and F.D.S. managed the survey program. M.M. and F.D.S. processed and organized the data sets and wrote the paper. MM conceived and coordinated the activities.

**Competing interests:** The authors declare no competing financial interests.


## Acknowledgements

The monitoring stations were settled in the frame of the Italian Flagship Project RITMARE (Italian Research for the Sea, where we operated as the research unit Co.N.I.S.Ma.-Polytechnic University of Bari; LIC- Coastal Engineering Laboratory) and with funds from PON R&C 2007-13 Project (chief scientist of both Michele Mossa). The present work was partially
funded by an agreement between the Polytechnic University of Bari and the Special Commissioner for urgent measures of reclamation, environmental improvements, and redevelopment of Taranto, gratefully acknowledged.

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
