# Peer review of "Meteo and hydrodynamic data in the Mar Grande and Mar Piccolo by the LIC Survey, winter and summer 2015"

_Earth System Science Data, 2020_

## Editor Comment (EC1) · Giuseppe M.R. Manzella (Editor) · 14 Oct 2020

Two important elements should be included in the paper: 1) Data formats. There are many data models and data formats (e.g. https://www.seadatanet.org/content/download/636/file/SDN2_D85_WP8_Datafile_formats.pdf - but also netCDF in general, oceanSites, etc). The authors should discuss their choices. 2) In 2010 a new standard for the properties of seawater called the thermo-dynamic equation of seawater 2010 (TEOS-10) was introduced, advocating absolute salinity as a replacement for practical salinity, and conservative temperature as a replacement for potential temperature. (https://en.wikipedia.org/wiki/Salinity). The

authors should discuss the common vocacularies generally adopted for the naming conventions (see e.g. https://vocab.seadatanet.org/search vocabularies P01, P02, P03)

---

## Author Comment (AC1) · 22 Oct 2020

Dear Editor, first of all we would like to thank you for the careful reading of our paper. Secondly, we appreciated the criticisms and the requests of clarification and integration, which made us possible to better explain our paper. We have reviewed our work according to your questions and, in the following, you will find a detailed answer to each of them.

Topical Editor Initial Decision: Start review and discussion after minor revisions (review by editor) (09 Sep 2020) by Giuseppe M.R. Manzella Comments to the Author:

[Figure]

Two important elements should be included in the paper: 1) Data formats. There are many data models and data formats (e.g. https://www.seadatanet.org/content/download/636/file/SDN2_D85_WP8_Datafile_formats.pdf) - but also netCDF in general, oceanSites, etc). The authors should discuss their choices.

Following this comment, in the revised version of paper, data format description has been added.

2) In 2010 a new standard for the properties of seawater called the thermo- dynamic equation of seawater 2010 (TEOS-10) was introduced, advocating absolute salinity as a replacement for practical salinity, and conservative temperature as a replacement for potential temperature. (https://en.wikipedia.org/wiki/Salinity). The authors should discuss the common vocabularies generally adopted for the naming conventions (see e.g. https://vocab.seadatanet.org/search vocabularies P01, P02, P03).

We agree with Editor's observation. In fact, in June 2009, a new Thermodynamic Equation of State of Seawater, referred to as TEOS-10, was adopted by the Scientific Committee on Oceanic Research (SCOR) and the International Association of Physical Sciences of the Ocean (IAPSO) Working Group 127 (WG127) (McDougall et al., 2009A). The new equation incorporates a more accurate representation of salinity known as Absolute Salinity. The main justification for preferring Absolute Salinity over Practical Salinity is that seawater's thermodynamic properties are directly influenced by the total mass of dissolved constituents (Absolute Salinity). However, the mass of dissolved constituents is regionally variable and are not always accurately represented when using conductivity measurements of seawater, the key parameter in the calculation of Practical Salinity. An algorithm is available that allows an estimate of Absolute Salinity to be expressed in terms of Practical Salinity (McDougall et al., 2009B).

The WG127 (SCOR/IAPSO Working Group 127 [2005 - 2012]) concluded there are very good reasons for continuing to store Practical Salinity rather than Absolute Salin-

ity in [such] data repositories: 1) Practical Salinity is an (almost) directly measured quantity whereas Absolute Salinity (the mass fraction of sea salt in seawater) is generally a derived quantity. (McDougall et al., 2009A). 2) it is imperative that confusion is not created in national data bases where there is a storing Absolute Salinity.

In the revised version of paper, the type of salinity and Temperature has been specified. In particular, our dataset contains a storing Practical Salinity (PSU) using PSS-78 algorithm and a storing Potential Temperature measured in ITS-90 degrees Celsius (°C).

McDougall, T.J., Feistel, R., Millero, F.J., Jackett, D.R., Wright, D.G., King, B.A., Marion, G.M., Chen, C-T.A., and Spitzer, P. 2009. Calculation of the Thermophysical Properties of Seawater,Global Ship-based Repeat Hydrography Manual, IOCCP Report No. 14, ICPO Publication Series no. 134. McDougall, R., Jackett, D.R., and Millero, F.J. 2009. An algorithm for estimating Absolute Salinity in the global ocean, Ocean Science Discussions,http://www.ocean-sci-discuss.net/6/215/2009/osd-6-215-2009.pdf

---

## Referee Comment (RC1) · Athanasia Iona (Referee) · 23 Oct 2020

SPECIFIC COMMENTS

(1) The monitoring stations described in this work were settled in the frame of the Italian Flagship Project RITMARE. One of the major aims of the RITMARE Project was the development of an interoperable Infrastructure for marine research. However, scientists usually disregard standards such as common terms for metadata, for parameters, data formats or controlled vocabularies as we can see in this data. The idea behind RITMARE was to provide scientists with tools and applications that would help scientists to share their data with a standardized and harmonized way. Is this monitoring activity addressing the RITMARE objectives? And if yes how?

(2) Page 5, line 109: The dataset is made of text files not excel files. Please correct it. Also correct it also at the (Data citation 1).

(3) Page 6, line 199: the format description needs some improvement. Explain what the first rows are about. Explain that the data are given in tab-separated columns, having a free text labeling (as it is given in the paper)

In each data file format description, use the same line separator and not only for the first one (e.g. the semicolon ; at the first row)

(4) Page 7: section 4 title: I would rather prefer to change it as "Technical and Data Validation"

(5) Page 7, lines 173-176: Data values outside the expected broad ranges are not always wrong values. Valuable information regarding the quality water changes caused by an extreme event either natural or human induced such as an accident, can be lost when eliminating instead of flagging data beyond plausible ranges.

A common practice in marine and ocean data management is the assignment of quality flags for each measurement. There are several quality flags schemas such as SeaDataNet, Ocean Data View, OceanSites, etc. I would suggest the use of such schemes in future releases of the data sets as well as the use of a common data format with standardized terms for metadata, parameters. This would facilitate the better understanding of data and their exchange with other research groups.

(6) Page 7, lines 177-179: the data validation description is quite general. As good data depends on good quality checks, the authors could provide more information on the conducted quality control checks with some examples if possible.

(7) Page 8, line 191: average values of dissolved oxygen and chlorophyll are not shown at Fig. 6.

(8) Pages 8, 9: the legends at Figures 3 are of smaller fond than the Figure 4, please homogenize.

(9) Page 10: Figure 5, there is a mismatch between y-axis label and legend (for temperature). Please correct.

Same for Figure 6, mismatches between x axis, y axis and legends. Also, correct the units of chlorophyll at the legend.

For these two figures, I would change the x-axis title. A suggestion could be: Time series of measured water temperature and salinity in July 2015. Horizontal time axis is in (month/day/year).

(10) Page 10, Authors Contributions: MM does not exist.

COMMENTS ON DATA FILES

(11) Essential metadata are missing from the data files such as the stations location (latitude and longitude) which makes impossible the geographical representation of the stations by plotting tools. I would suggest to add them, in this way the data could be easily be plotted by tools like ODV together with the geographical positions of the measuring stations.

(12) In MP-TA-01-2015-temperture.txt file: the header description (line 3) says WATER QUALITY instead of WATER TEMPERATURE.

Same for MP-TA-07-2015-temperature.txt data file. Needs correction.

(13) The MG001, 2_meteo.txt data sets do not include a relevant header description as the rest of the data sets

---

## Author Comment (AC2) · 28 Oct 2020

Athanasia Iona (Referee) sissy@hnodc.hcmr.gr

GENERAL COMMENTS The paper describes two monthly coastal data sets of 2015, collected by two fixed stations at Mar Grande and Mar Piccolo within the frame of a monitoring activity. The generation of times series data in sensitive and vulnerable areas as in this study, are of high importance for understanding the hydrodynamic structures and characteristics of the area and for supporting the coastal management. In

addition, such monitoring data can actively contribute to the successful implementation of national policies and priorities such the MSFD. Unless there are data management processes not mentioned in this work that are undertaken by data centres at more central national level, more standardization at future releases of the data sets would be beneficial. Some modifications, from my point of view, are needed before publication. See attached supplement for more details. Please also note the supplement to this comment: https://essd.copernicus.org/preprints/essd-2020-229/essd-2020-229-RC1- supplement.pdf

We would like to thank Reviewer for her careful reading of our paper and comments. We appreciated criticisms and requests of clarification and integration, which allowed us to better explain our paper. We have reviewed our work accordingly, and detailed answers are shown in the following.

(1) The monitoring stations described in this work were settled in the frame of the Italian Flagship Project RITMARE. One of the major aims of the RITMARE Project was the development of an interoperable Infrastructure for marine research. However, scientists usually disregard standards such as common terms for metadata, for parameters, data formats or controlled vocabularies as we can see in this data. The idea behind RITMARE was to provide scientists with tools and applications that would help scientists to share their data with a standardized and harmonized way. Is this monitoring activity addressing the RITMARE objectives? And if yes how?

Really, the monitoring project was funded by the Italian National Project PON R&C 2007-2013 "Magna Grecia" and (also) by the RITMARE flagship project. Obviously, the aim of the paper is not to report on the activities of RITMARE, where interested readers can have further details from the website http://www.ritmare.it/

Another source of RITMARE, at least as regards the working group made up of the authors of this paper, are the papers, many of which are referred to in the paper itself. Others will be inserted in the revised manuscript.

[Figure]

(2) Page 5, line 109: The dataset is made of text files not excel files. Please correct it. Also correct it also at the (Data citation 1).

The correction will be added in the revised manuscript.

(3) Page 6, line 199: the format description needs some improvement. Explain what the first rows are about. Explain that the data are given in tab-separated columns, having a free text labeling (as it is given in the paper) In each data file format description, use the same line separator and not only for the first one (e.g. the semicolon ; at the first row)

Following this comment, in the revised version of paper, data format description will be added.

(4) Page 7: section 4 title: I would rather prefer to change it as "Technical and Data Validation"

Following this comment, in the revised version of paper, the section 4 title will be changed.

(5) Page 7, lines 173-176: Data values outside the expected broad ranges are not always wrong values. Valuable information regarding the quality water changes caused by an extreme event either natural or human induced such as an accident, can be lost when eliminating instead of flagging data beyond plausible ranges. A common practice in marine and ocean data management is the assignment of quality flags for each measurement. There are several quality flags schemas such as SeaDataNet, Ocean Data View, OceanSites, etc. I would suggest the use of such schemes in future releases of the data sets as well as the use of a common data format with standardized terms for metadata, parameters. This would facilitate the better understanding of data and their exchange with other research groups.

We agree with this comment and advice.

(6) Page 7, lines 177-179: the data validation description is quite general. As good
data depends on good quality checks, the authors could provide more information on the conducted quality control checks with some examples if possible.

Maintenance and calibration of instruments occur twice a year in specialized laboratories and using set of measured data during monitoring survey using Vessel-Mounted Instruments such as Nortek AWAC Vessel Mounted Acoustic Doppler Current Profiler (VM-ADCP)

(7) Page 8, line 191: average values of dissolved oxygen and chlorophyll are not shown at Fig. 6.

Reviewer is right, being the sentences not clear. We will rephrase in the new version of the manuscript.

(8) Pages 8, 9: the legends at Figures 3 are of smaller fond than the Figure 4, please homogenize.

The legends are different because the plotted items of Figure3 and 4 are different. Figure 3 shows the pattern of the bottom and surface currents instead Figure 4 shows measured wind and waves. However, the legends can be changed if necessary.

(9) Page 10: Figure 5, there is a mismatch between y-axis label and legend (for temperature). Please correct. Same for Figure 6, mismatches between x axis, y axis and legends. Also, correct the units of chlorophyll at the legend. For these two figures, I would change the x-axis title. A suggestion could be: Time series of measured water temperature and salinity in July 2015. Horizontal time axis is in (month/day/year).

The corrections will be added in the revised manuscript

(10) Page 10, Authors Contributions: MM does not exist.

MM is for M.M. However, the correction will be added in the revised manuscript.

COMMENTS ON DATA FILES (11) Essential metadata are missing from the data files such as the stations location (latitude and longitude) which makes impossible the geographical representation of the stations by plotting tools. I would suggest adding them, in this way the data could be easily be plotted by tools like ODV together with the geographical positions of the measuring stations.

The data files are tab-delimited text format (ASCII), and the stations location (latitude and longitude) is indicated in section 2. However, this information will be added also in text files in the revised paper.

(12) In MP-TA-01-2015-temperture.txt file: the header description (line 3) says WATER QUALITY instead of WATER TEMPERATURE.

The correction will be added in the MP-TA-01-2015-temperture.txt file

Same for MP-TA-07-2015-temperature.txt data file. Needs correction.

The corrections will be added in both files.

(13) The MG001, 2_meteo.txt data sets do not include a relevant header description as the rest of the data sets

The corrections will be added in the file.
* * *

---

## Referee Comment (RC2) · Stefano Sibilla (Referee) · 4 Nov 2020

General comments

The paper describes the meteorologic and hydrodynamic datasets collected at two fixed stations located in the Mar Grande (MG) and in the Mar Piccolo (MP) bays of the Taranto harbour. Both stations are equipped with an ADCP profiler and a wave array; the first station is also equipped with a weather station and sensors of different water quality parameters. The stations allowed the Authors to collect, for a winter and a summer month in 2015, current profile, wave, meteo and water quality data records at the MG station and current profile, wave and temperature data records at the MP

station.

The provided datasets can be used to both complement and validate the findings of other studies and numerical models, in order to better understand the hydrodynamic and biological patterns which characterize this complex coastal basin. The good quality and completeness of these datasets can therefore highly contribute to transform the Mar Grande/Mar Piccolo case study in a benchmark case, upon which methods and models can be tested, before extending them to similar basins.

Specific comments

In the paper, the Authors show a sample of the results obtained from the MG station in terms of direction and intensity of surface and bottom currents, of winds and waves, as well as time series of temperature, salinity, dissolved oxygen and chlorophyll. A brief interpretation of these data, in terms of current circulation patterns and of water quality time trends, is also proposed.

The results obtained from the MP station are instead not shown and commented. I understand that this is probably due to conciseness reasons; however, a brief comment about the quality and use of these data may add value to the paper. In particular, the Authors may explain why the MP station is located in the navigable channel between the two bays and which are the advantages (or the limits, if any) of this location in terms of information given by the station on the circulation regimes in the Mar Piccolo and on the flows exchanged through the channel itself.

———————————————

---

## Author Comment (AC3) · 9 Nov 2020

Stefano Sibilla (Referee) stefano.sibilla@unipv.it

GENERAL COMMENTS

The paper describes the meteorologic and hydrodynamic datasets collected at two fixed stations located in the Mar Grande (MG) and in the Mar Piccolo (MP) bays of

the Taranto harbour. Both stations are equipped with an ADCP profiler and a wave array; the first station is also equipped with a weather station and sensors of different water quality parameters. The stations allowed the Authors to collect, for a winter and a summer month in 2015, current profile, wave, meteo and water quality data records at the MG station and current profile, wave and temperature data records at the MP station. The provided datasets can be used to both complement and validate the findings of other studies and numerical models, in order to better understand the hydrodynamic And biological patterns which characterize this complex coastal basin. The good quality and completeness of these datasets can therefore highly contribute to transform the Mar Grande/Mar Piccolo case study in a benchmark case, upon which methods and models can be tested, before extending them to similar basins.

We would like to thank the Reviewer for his careful reading of our paper and comments. We appreciated criticisms and requests of clarification and integration, which allowed us to better explain our paper. We will review our work accordingly, and detailed answers are shown in the following.

Specific comments

In the paper, the Authors show a sample of the results obtained from the MG station in terms of direction and intensity of surface and bottom currents, of winds and waves, as well as time series of temperature, salinity, dissolved oxygen and chlorophyll. A brief interpretation of these data, in terms of current circulation patterns and of water quality time trends, is also proposed. The results obtained from the MP station are instead not shown and commented. I understand that this is probably due to conciseness reasons; however, a brief comment about the quality and use of these data may add value to the paper.

Following this comment, in the revised version of paper, a sample of the results obtained from the MP station will be added.

Authors may explain why the MP station is located in the navigable channel between

the two bays and which are the advantages (or the limits, if any) of this location in terms of information given by the station on the circulation regimes in the Mar Piccolo and on the flows exchanged through the channel itself.

The Mar Piccolo site with its typical lagoon features, is extremely vulnerable and is often suffering scarce circulation. The main exchange flow is through the Navigable Channel. As shown by previous studies and field data analysis from MP station (Armenio et al., 2017; De Serio and Mossa, 2018; De Serio et al., 2020) the net flow is inflowing at deeper layers, from the bottom up to 4m depth, while in the most superficial layer it is directed outward of the basin. Therefore, monitoring actions, as that of MP station, is necessary to better understand the most significant hydrodynamic–biological variability of this extremely vulnerable coastal site. The results of this study can be applied for similar zones. Following Referee's advice, this information will be added also in text files in the revised paper.

Armenio, E., De Serio, F., Mossa, M. Analysis of data characterizing tide and current fluxes in coastal basins. Hydrology and Earth System Sciences, 21 (7), 3441-3454, 2017.

De Serio, F., Mossa, M. Meteo and hydrodynamic measurements to detect physical processes in confined shallow seas. Sensors 18 (1), 280, 2018.

De Serio, F, Armenio, E., Ben Meftah, M., Capasso, G., Corbelli, V., De Padova, D., De Pascalis, F., Di Bernardino, A., Leuzzi, G., Monti, P., Pini, A., Velardo, R., Mossa, M. Detecting sensitive areas in confined shallow basins. ENVIRONMENTAL MODELLING & SOFTWARE, vol. 126, ISSN: 1364-8152, doi: 10.1016/j.envsoft.2020.104659, 2020.

---

## Editor Decision (ED1)

Comment to:

Meteo and hydrodynamic data in the Mar Grande and Mar Piccolo by the LIC Survey, winter and summer 2015

by Michele Mossa, Elvira Armenio, Mouldi Ben Meftah, Maria Francesca Bruno, Diana De Padova, Francesca De Serio

General comment: The paper has been improved, but not all the referees' questions have been addressed. The authors are invited to answer to questions and also review the english.

Specific comments:
1) The paragraph 3 is not easy to read. I suggest to synthesise the information in a unique table such as:

| | | Mare Piccolo | Mare Grande |
|---|---|---|---|
| waves | Progressive data number | | |
| | Date (year/month/day/hour/minute) | x | x |
| | Position ??? | Significant wave height Hs
Significant wave period Ts
Significant wave incoming direction (in degree, referenced to North)
Average of the 1/10 highest waves H1/10 (m)
Average wave period Tmean (s). | |
| Currents | Progressive data number | | |
| | Date (year/month/day/hour/minute) | x | x |
| | Position ??? | Cell of measurement with indication of its depth from surface (z=0);
cell current intensity (m/s)
cell current direction | |
| Temp. | Progressive data number | | |
| | Date (year/month/day/hour/minute) | x | |
| | Position??? | Water Potential Temperature measured in ITS-90 degrees Celsius (°C) | |
| Meteo | Progressive data number | | |
| | Date (year/month/day/hour/minute) | | x |
| | Position ??? | Average wind velocity (m/s) ;
Max wind velocity (m/s);
Wind incoming direction NE (deg);
Air temperature (°C); | |

| Wat Qual | | | |
|---|---|---|---|
| | | | Dew point (°C);

Atmospheric pressure (mbar);

Relative humidity (%). |
| | Progressive data number | | |
| | Date (year/month/day/hour/minute) | | x |
| | Position??? | | Water Potential Temperature measured in ITS-90 degrees Celsius (°C);

Conductivity (S/m);
Pressure (dbar);

Practical Salinity (PSU) using PSS-78 algorithm
Density (kg/m3);

Dissolved oxygen (ml/l);

Chlorophyll (µg/l):
Turbidity (NTU)
CDOM (RFU)
Crude oil (RFU)
Refined oil (RFU) |

Referee Iona in 'comments on data files' suggest to add also 'position' in the data files. This information must be in the table. The referee, in practice, ask for interoperability, i.e. the possibility to exchange data following international agreed metadata content and data formats (Oceav Data View is cited).

2) The other important referee comment is the n.5. Here is a request to use common vocabularies (e.g. the SeaDataNet common vocabularies such as the P02). These vocabularies must be used in the table, and must be presented in the text.

3) Flags and quality control: the authors are presenting succinctly the quality assurance procedures, but the quality control is lacking. Also, in this case, the authors could refer to SeaDataNet Quality control. It must also be said how the data were flagged.

The authors are asked to answer to all Iona questions and include their answers in the text.

Giuseppe M.R. Manzella

---

## Author Response (AR2)

Dear Editor,
first of all we would like to thank you for the careful reading of our paper.
Secondly, we appreciated the criticisms and the requests of clarification and integration, which made us possible to better explain our paper.
We have reviewed our work according to your questions and, in the following, you will find a detailed answer to each of them.

**Meteo and hydrodynamic data in the Mar Grande and Mar Piccolo by the LIC Survey, winter and summer 2015**

**by Michele Mossa, Elvira Armenio, Mouldi Ben Meftah, Maria Francesca Bruno, Diana De Padova, Francesca De Serio**

**General comment: The paper has been improved, but not all the referees' questions have been addressed. The authors are invited to answer to questions and also review the english.**

**Specific comments:**

1) **The paragraph 3 is not easy to read. I suggest to synthesise the information in a unique table such as:**

| | | Mare Piccolo | Mare Grande |
|---|---|---|---|
| waves | Progressive data number | | |
| | Date (year/month/day/hour/minute) | x | x |
| | Position ??? | Significant wave height Hs Significant wave period Ts Significant wave incoming direction (in degree, referenced to North) Average of the 1/10 highest waves H1/10 (m) Average wave period Tmean (s). | |
| Currents | Progressive data number | | |
| | Date (year/month/day/hour/minute) | x | x |
| | Position ??? | Cell of measurement with indication of its depth from surface (z=0); cell current intensity (m/s) cell current direction | |
| Temp. | Progressive data number | | |
| | Date (year/month/day/hour/minute) | x | |
| | Position??? | Water Potential Temperature measured in ITS-90 degrees Celsius (°C) | |
| Meteo | Progressive data number | | |
| | Date (year/month/day/hour/minute) | | x |
| | Position ??? | Average wind velocity (m/s) ; Max wind velocity (m/s); | |

| | | | |
|---|---|---|---|
| | | Wind incoming direction NE (deg); | |
| | | Air temperature (°C); | |
| | | Dew point (°C); | |
| | | Atmospheric pressure (mbar); | |
| | | Relative humidity (%). | |
| Wat Qual | Progressive data number | | |
| | Date (year/month/day/hour/minute) | | x |
| | Position??? | Water Potential Temperature measured in ITS-90 degrees Celsius (°C); | |
| | | Conductivity (S/m); | |
| | | Pressure (dbar); | |
| | | Practical Salinity (PSU) using PSS-78 algorithm | |
| | | Density (kg/m3); | |
| | | Dissolved oxygen (ml/l); | |
| | | Chlorophyll (μg/l): | |
| | | Turbidity (NTU) | |
| | | CDOM (RFU) | |
| | | Crude oil (RFU) | |
| | | Refined oil (RFU) | |

**Referee Iona in 'comments on data files' suggest to add also 'position' in the data files. This information must be in the table. The referee, in practice, ask for interoperability, i.e. the possibility to exchange data following international agreed metadata content and data formats (Oceav Data View is cited).**

**2) The other important referee comment is the n.5. Here is a request to use common vocabularies (e.g. the SeaDataNet common vocabularies such as the P02). These vocabularies must be used in the table, and must be presented in the text.**

*We here address these two questions altogether.*
*Following these comments and Iona advice, the paragraph 3 has been improved in the revised version of paper. Moreover, the meteocean variables have been marked up with the SeaDataNet common vocabularies from Library P01, P02 and P0 in the revised version of paper.*

*The dataset supplied in tab-delimited text format ASCII, contains timeseries of relevant meteocean variables marked up with the SeaDataNet common vocabularies from Library P01, P02 and P03 (https://vocab.seadatanet.org/search vocabularies P01, P02, P03) and divided as follows:*

| | | Mare Piccolo (MP) | Mare Grande (MG) |
|---|---|---|---|
| **Wave** | Progressive data number
Date (year/month/day/hour/minute)
Position (Lat, lon) | x | x |
| | | SDN: P01: Conceptid:: GTHDAP01: Significant wave height $H_s$(m)
- SDN: P01: Conceptid:: GTZHAW01: Significant wave period $T_s$ (s);
- SDN: P01: Conceptid:: GWMDAD01: Significant wave incoming direction
(in degree, referenced to North);
- SDN: P01: Conceptid:: MBANZZZZ:Local depth (mm);
- SDN: P01: Conceptid:: GTDHAP01: $H_{1/10}$- Average of the 1/10 highest waves;
- SDN: P01: Conceptid:: GTAMZD01: Average wave period $T_{mean}$ (s) | |
| **Current** | Progressive data number
Date (year/month/day/hour/minute)
Position (Lat, lon) | x | x |
| | | - SDN: P01: Conceptid:: MBANZZZZ: Cell of measurement with indication of its depth from surface (z=0);
- SDN: P01: Conceptid:: LCSAAP01: cell current intensity (m/s);
- SDN: P01: Conceptid:: LCDAAP01: cell current direction (in degree, referenced to North) | |
| **Temperature** | Progressive data number
Date (year/month/day/hour/minute)
Position (Lat, lon) | x | |
| | | - SDN: P01: Conceptid:: MBANZZZZ: Sensor depth (m);
- SDN: P01: Conceptid:: TEMPS901: Water Potential Temperature measured in ITS-90 degrees Celsius (°C) | |
| **Meteo** | Progressive data number
Date (year/month/day/hour/minute)
Position (Lat, lon) | | x |
| | | - SDN: P01: Conceptid:: EGTSSS01: Average wind velocity (m/s) ;
- SDN: P01: Conceptid:: ESSAMX01: Max wind velocity (m/s);
- SDN: P01: Conceptid:: EGTDSS01: Wind incoming direction N (deg);
- SDN: P01: Conceptid:: CDTAZZ01: Air temperature (°C);
- SDN: P01: Conceptid:: CDEWZZ01:Dew point (°C);
- SDN: P01: Conceptid:: CAPHZZ01:Atmospheric pressure (mbar);
- SDN: P01: Conceptid:: CHUMZZ01: Relative humidity (%). | |
| **Water quality** | Progressive data number
Date (year/month/day/hour/minute) | | x |
| | | - SDN: P01: Conceptid:: TEMPS901: Water Potential Temperature measured in ITS-90 degrees Celsius | |

| | | |
|---|---|---|
| | Position (Lat, lon) | (°C);
- SDN: P01:Conceptid:: CNDCST01: Conductivity (S/m);
- SDN: P01: Conceptid:: PRESPR01: Absolute Pressure (dbar);
- SDN: P01 :Conceptid:: PSLTZZ01: Practical Salinity (PSU) using PSS78 algorithm;
- SDN: P01:Conceptid:: SIGTPR01: Density (kg/m$^3$);
- SDN: P01: Conceptid:: DOXYOP01:Dissolved oxygen (ml/l);
- SDN: P01: Conceptid:: CLSDPM01: Chlorophyll (µg/l);
- SDN: P01: Conceptid:: CLSDPM01: Turbidity (NTU);
- SDN: P01: Conceptid:: GP001:CDOM (RFU);
- SDN: P01: Conceptid:: GP001: Crude oil (RFU);
- SDN: P01: Conceptid:: GP001: Refined oil (RFU). |

| | | Mare Piccolo (MP) | Mare Grande (MG) |
|---|---|---|---|
| **Wave** | Progressive data number
Date (year/month/day/hour/minute)
Position (Lat, lon) | x | x |
| | | SDN: P01: Conceptid:: GTHDAP01: Significant wave height $H_s$(m)
- SDN: P01: Conceptid:: GTZHAW01: Significant wave period $T_s$ (s);
- SDN: P01: Conceptid:: GWMDAD01: Significant wave incoming direction
(in degree, referenced to North);
- SDN: P01: Conceptid:: MBANZZZZ:Local depth (mm);
- SDN: P01: Conceptid:: GTDHAP01: $H_{1/10}$- Average of the 1/10 highest waves;
- SDN: P01: Conceptid:: GTAMZD01: Average wave period $T_{mean}$ (s) | |

**3) Flags and quality control: the authors are presenting succinctly the quality assurance procedures, but the quality control is lacking. Also, in this case, the authors could refer to SeaDataNet Quality control. It must also be said how the data were flagged.**
**The authors are asked to answer to all Iona questions and include their answers in the text.**

*Following this comment, a description of Quality Control has been added in the revised paper.*

*The data set has been processed with quality control procedures and data flagged following SeaDataNet protocols. In particular, the dataset quality control has been carried out with:*
*-      Maintenance and calibration of instruments twice a year in specialized laboratories;*

- *Visual inspection of the time series (e.g. time series plot, current vector scatter plot, progressive vector diagram, etc.);*
- *Screening together of related parameters such as current speed and current direction or salinity and temperature to identify spurious values;*
- *Flag spikes in the data;*
- *Flag suspicious data or correct the data after consultation with the data supplier;*
- *Check against other data collected on nearby moorings or measured during monitoring survey using two Vessel-Mounted Acoustic Doppler Current Profilers (VM-ADCPs).*

*Best regards*

*Michele Mossa*
*Elvira Armenio,*
*Mouldi Ben Meftah*
*Maria Francesca Bruno*
*Diana De Padova*
*Francesca De Serio*